# ESM-1 Overexpression is Involved in Increased Tumorigenesis of Radiotherapy-Resistant Breast Cancer Cells

**DOI:** 10.3390/cancers12061363

**Published:** 2020-05-26

**Authors:** Hana Jin, Trojan Rugira, Young Shin Ko, Sang Won Park, Seung Pil Yun, Hye Jung Kim

**Affiliations:** 1Department of Pharmacology, College of Medicine, Institute of Health Sciences, Gyeongsang National University, Jinju 52727, Korea; hanajin.kr@daum.net (H.J.); rugirawacu@gmail.com (T.R.); shini33@naver.com (Y.S.K.); parksw@gnu.ac.kr (S.W.P.); spyun@gnu.ac.kr (S.P.Y.); 2Department of Convergence Medical Science (BK21 Plus), Gyeongsang National University, Jinju 52727, Korea

**Keywords:** ESM-1, metastasis, tumorigenesis, radiotherapy-resistant, TNBC

## Abstract

The key barrier to the effectiveness of radiotherapy remains the radioresistance of breast cancer cells, resulting in increased tumor recurrence and metastasis. Thus, in this study, we aimed to clarify the difference between radiotherapy-resistant (RT-R) breast cancer (BC) and BC, and accordingly, analyzed gene expression levels between radiotherapy-resistant (RT-R) MDA-MB-231 cells and MDA-MB-231 cells. Gene expression array showed that ESM-1 was the most upregulated in RT-R-MDA-MB-231 cells compared to MDA-MB-231 cells. Then, we aimed to investigate the role of ESM-1 in the increased tumorigenesis of RT-R-BC cells. RT-R-MDA-MB-231, which showed an increased expression level of ESM1, exhibited significantly enhanced proliferation, colony forming ability, migration, and invasion compared to MDA-MB-231 cells, and ESM-1 knockdown effectively reversed these effects. In addition, compared to MDA-MB-231 cells, RT-R-MDA-MB-231 cells displayed improved adhesion to endothelial cells (ECs) due to the induction of adhesion molecules and increased MMP-9 activity and VEGF-A production, which were decreased by ESM-1 knockdown. Moreover, the expression of HIF-1α and activation of NF-κB and STAT-3 were increased in RT-R-MDA-MB-231 cells compared to MDA-MB-231 cells, and these effects were abolished by the knockdown of ESM-1. Finally, we confirmed the role of ESM-1 in tumorigenesis in an in vivo mouse model. Tumor volume, lung metastasis, and tumorigenic molecules (VEGF-A, HIF-1α, MMP-9, ICAM-1, VCAM-1, and phospho-NF-κB and phospho-STAT-3) were significantly induced in mice injected with ESM-1-overexpressing 4T1 cells and greatly enhanced in those injected with ESM-1-overexpressing RT-R-4T1 cells. Taken together, these results suggest for the first time that ESM-1 plays a critical role in tumorigenesis of breast cancer cells, especially RT-R-breast cancer cells, through the induction of cell proliferation and invasion.

## 1. Introduction

Breast cancer (BC) is the most common cancer diagnosed in women worldwide. BC incidence is relatively high in North American and Western European women, but the incidence and mortality rates in Asian populations and low-income countries have been steadily increasing due to a lack of diagnostic and treatment programs [1,2]. BC is a diverse disease consisting of many subtypes, each of which carries a different molecular and pathologic profile. Triple negative breast cancer (TNBC) is a subtype of BC characterized by a lack of estrogen and progesterone receptors (ER/PR) expression and epidermal growth factor receptor 2 (HER2) expression. Epidemiologically, TNBC accounts for 12% to 18% of all BC cases. TNBC patients do not benefit from hormonal therapy due to the lack of receptor expression. Even though therapeutic methods including surgery, chemotherapy, and radiotherapy have been developed to treat cancer, TNBC patients easily develop therapeutic resistance and disease recurrence [3]. Radiotherapy (RT) is a method mostly used to treat solid cancers. Although RT has improved the lives of BC patients, some patients still develop tumor recurrence after completion of RT. Tumor recurrence after RT may be caused by aggressive tumor biology, the capacity of cancer cells to resist RT, such as the presence of cancer stem cells (CSCs) and hypoxic conditions, or the development of de novo resistance, leading to treatment failure [4].

In our previous study [5], we established radiotherapy-resistant (RT-R)-BC cells from TNBC cell (MDA-MB-231) and nonTNBC cells (MCF-8 and T47D) by repeated irradiation 25 times at 2 Gy to accumulate total dose of 50 Gy, and then examined the properties of RT-R-BC cells. We found that RT-R-MDA-MB-231 cells, which are derived from highly metastatic MDA-MB-231 cells, showed most radio- and chemo-resistance of tested three cell lines (RT-R-MDA-MB-231, RT-R-MCF-7, and RT-R-T47D) and more increased protein levels of CSCs markers CD44, Notch-4, Oct3/4, and ALDH1, compared to other RT-R-BC cells. Moreover, compared to MDA-MB-231 cells, RT-R-MDA-MB-231 cells exhibited increased metastatic properties due to higher expression of ICAM-1 and VCAM-1, which promoted migration and adhesion to endothelial cells (ECs) and induced invasion through ECs by increasing MMP-9, Snail-1, and β-catenin production and decreasing E-cadherin [5]. Furthermore, we reported that RT-R-MDA-MB-231 cells produced more CSCs, and suggested CSCs may be the source of radioresistance and RT acquisition [5]. In addition, we have also showed that RT-R-MDA-MB-231 cells release higher levels of ATP than other BC cells, and the subsequent activation of P2Y purinergic receptor 2 (P2Y_2_R) by released ATP leads to tumor growth and invasion through inflammasome activation [6]. Therefore, in this study, we aimed to clarify the differential gene expression level between RT-R-MDA-MB-231 cells and MDA-MB-231 cells, which showed most radioresistance and metastatic properties and obtained the result showing that endothelial cell specific molecule-1 (ESM-1) was most upregulated in RT-R-MDA-MB-231 cells than in MDA-MB-231 cells.

ESM-1, also called endocan, is a 50-kDa secreted proteoglycan. ESM-1 is composed of a 165-amino acid mature polypeptide and one dermatan sulfate chain [7] and is secreted by diverse ECs [8]. ESM-1 secretion is upregulated by angiogenic factors such as VEGF or proinflammatory cytokines such as tumor necrosis factor (TNFα), interleukin (IL)-1β, and IFNγ [9,10,11,12]. Moreover, a higher concentration of ESM-1 has been detected in the serum of patients with septic shock and sepsis than in the serum of healthy people [10,13]. Furthermore, a variety of studies have reported that a high level of ESM-1 secretion is found in several cancers, including lung cancer [14], kidney cancer [15,16], colorectal cancer [17], acute myeloid leukemia [18], glioblastomas [7], and BC [19]. Moreover, ESM-1 overexpression in human embryonic kidney 293 cells causes tumor formation and growth in SCID mice [20]. However, the role of ESM-1 in enhanced tumorigenesis in RT-R TNBC has not yet been reported. Therefore, in this study, we also aimed to investigate the role of ESM-1 overexpression in the tumorigenesis of RT-R-MDA-MB-231 BC cells.

## 2. Results

### 2.1. ESM-1 is Significantly Overexpressed in RT-R-MDA-MB-231 Cells Compared with MDA-MB-231 Cells

First, we analyzed gene expression levels between MDA-MB-231 and RT-R-MDA-MB-231 cells by gene expression array analysis, and we found that the ESM-1 mRNA level was most upregulated in RT-R-MDA-MB-231 cells than in MDA-MB-231 cells (Table 1). Then, we confirmed the expression levels of ESM-1 in MDA-MB-231 cells and RT-R-MDA-MB-231 cells in both mRNA level and protein level by RT-PCR and ELISA, respectively (Figure 1). The results confirmed that RT-R-MDA-MB-231 cells showed higher expression levels of ESM-1 than MDA-MB-231 cells at both the mRNA and protein levels.

### 2.2. ESM-1 Knockdown Inhibits the Proliferation, Colony Formation Ability, Migration, and Invasion of both MDA-MB-231 and RT-R-MDA-MB-231 Cells

Then, we investigated the effect of ESM-1 overexpression on the increased tumorigenesis of RT-R-MDA-MB-231 cells. First, we knocked down the ESM-1 gene in both MDA-MB-231 and RT-R-MDA-MB-231 cells by transfecting a small interfering RNA (siRNA) and confirmed the effectiveness of knockdown by RT-PCR (Figure 2A). Cell proliferation was evaluated by the Cell Counting Kit-8 (CCK-8) assay. RT-R-MDA-MB-231 cells demonstrated a higher cell proliferation ability (55%) than their parental MDA-MB-231 cells, and this effect was significantly reduced by ESM-1 knockdown (Figure 2B). In addition, RT-R-MDA-MB-231 cells showed significantly enhanced ability to form colony compared to MDA-MB-231 cells, and this ability was significantly reduced by ESM-1 knockdown (Figure 2C,D). According to the report [21], during tumor metastasis, cancer cells migrate from their primary sites to invade new distant organs. Therefore, to evaluate the role of ESM-1 in BC cell migration and invasion, ESM-1 was knocked down in both MDA-MB-231 and RT-R-MDA-MB-231 cells, and then cell migration and invasion were analyzed. Our results showed that the migration abilities of both MDA-MB-231 and RT-R-MDA-MB-231 cells transfected with ESM-1 siRNA were decreased more than 2-fold and 3-fold, compared to the migration abilities of those transfected with control (CTRL) siRNA, when they measured at 24 h and 48 h, respectively (Figure 2E,F). Additionally, the invasion of RT-R-MDA-MB-231 cells through the EC-Matrigel-coated insert well membrane was 2.25-fold greater than that of MDA-MB-231 cells, and the invasion of both RT-R-MDA-MB-231 and MDA-MB-231 cells transfected with ESM siRNA was significantly reduced (Figure 2G,H).

### 2.3. ESM-1 Plays a Role in Tumorigenesis in MDA-MB-231 and RT-R-MDA-MB-231 Cells through Induction of Adhesion Molecules, Resulting in Adhesion of these Cells to ECs, MMP-9 Activity, and VEGF-A Production

Cell adhesion molecules, such as ICAM-1 and VCAM-1, are involved in cancer growth, migration from primary sites to distant organs, and adhesion to ECs [22,23]. Accordingly, we examined the role of ESM-1 in cell adhesion molecule expression in MDA-MB-231 and RT-R-MDA-MB-231 cells and the subsequent adhesion of these cells to ECs. As shown in Figure 3A–C, ICAM-1 and VCAM-1 protein levels were increased in RT-R-MDA-MB-231 cells compared to MDA-MB-231 cells, and ESM-1 siRNA-transfected RT-R-MDA-MB-231 and MDA-MB-231 cells decreased ICAM-1 and VCAM-1 protein levels. Moreover, 1.7-fold more RT-R-MDA-MB-231 cells than MDA-MB-231 cells adhered to ECs, and the adhesion of ESM-1 siRNA-transfected MDA-MB-231 and RT-R-MDA-MB-231 cells to ECs was significantly decreased compared to that of MDA-MB-231 and RT-R-MDA-MB-231 cells transfected with CTRL siRNA, respectively (Figure 3D–F). During cancer invasion and metastasis, MMPs destroy the surrounding basement membrane, allowing cancer cells to spread to new tissues and inducing the formation of new blood vessels through a process called angiogenesis for tumor growth and persistence. Therefore, we analyzed the effect of ESM-1 on MMP-9 activity and VEGF-A production. As similar with the previous data, RT-R-MDA-MB-231 cells showed increased MMP-9 activity and VEGF-A production compared to that observed in MDA-MB-231 cells, and RT-R-MDA-MB-231 and MDA-MB-231 cells transfected with ESM-1 siRNA exhibited significantly decreased MMP-9 activity (Figure 3G) and VEGF-A production (Figure 3H).

### 2.4. ESM-1-Overexpressing RT-R-MDA-MB-231 Cells Increase ERK1/2, PKC, and PDK1 Activation and then Transcription Factor Hypoxia-Inducible Factor-1α (HIF-1α) Induction and NF-κB and STAT-3 Activation

Then, we investigated the effect of ESM1 on the intracellular signaling molecules, which are involved in cell survival, cell migration, and metastasis. We found that ERK1/2, PKC, and PDK1 phosphorylations were highly induced in RT-R-MDA-MB-231 cells than in MDA-MB-231 cells. Phosphorylated ERK1/2, PKC, and PDK1 protein levels were significantly suppressed in both MDA-MB-231 and RT-R-MDA-MB-231 cells transfected with ESM1 siRNA (Figure 4A–F). Based on the reports, these ERK1/2, PKC, and PDK1 pathways potentiate the activation of HIF-1α, NF-κB, and STAT-3, and these transcription factors are well known to be involved in inflammatory responses to promote cancer development and growth through the transcriptional regulation of various inflammatory genes, such as adhesion molecules, MMPs, and VEGF [24,25,26,27,28,29,30]. Thus, we wanted to determine the effect of ESM-1 overexpression on these transcription factors in RT-R-MDA-MB-231 cells. Western blot analysis revealed that compared with MDA-MB-231 cells, RT-R-MDA-MB-231 cells overexpressing ESM-1 exhibited enhanced HIF-1α protein expression and NF-κB and STAT-3 activation and that ESM-1 knockdown induced by siRNA reduced HIF-1α protein expression and NF-κB and STAT-3 activation in RT-R-MDA-MB-231 and MDA-MB-231 cells (Figure 5A–F).

### 2.5. ESM-1 Overexpression Increases Tumor Growth and Metastasis in an In Vivo Mouse Model

Finally, we confirmed the role of ESM-1 in tumorigenesis in an in vivo mouse model. We divided athymic nude mice into four groups and subcutaneously injected them with empty plasmid vector-transfected 4T1 (4T1) cells, empty plasmid vector-transfected RT-R-4T1 (RT-R-4T1) cells, ESM-1 plasmid vector-transfected 4T1 (4T1-ESM-1) cells, or ESM-1 plasmid vector-transfected RT-R-4T1 (RT-R-4T1-ESM-1) cells (Figure 6A). Tumor volume and body weight were measured every 3 days for 24 days. The tumor volume of mice injected with 4T1-ESM-1 cells and RT-R-4T1-ESM-1 cells was significantly increased from the 14th day after injection (Figure 6B). In addition, we found that mice injected with RT-R-4T1-ESM-1 cells presented the greater increase in tumor volume (Figure 6C,D), even though the body weights of the four groups were not significantly different during tumor development (Figure 6E). Interestingly, on the day that the mice were sacrificed, lung metastasis examination showed that RT-R-4T1 cell-injected mice showed an increased number of lung metastases compared to that exhibited by 4T1 cell-injected mice, and this effect was more prominent in mice injected with 4T1-ESM-1 cells and RT-R-4T1-ESM-1 cells; 4T1-ESM-1 cell-injected and RT-R-4T1-ESM-1 cell-injected mice showed a significant increase in lung metastasis compared to that of 4T1 cell- and RT-R-4T1 cell-injected mice, respectively (Figure 6F). The tumor angiogenic marker VEGF-A was significantly increased in mice injected with 4T1-ESM-1 and RT-R-4T1-ESM-1 cells (Figure 6G). Immunohistochemistry (IHC) staining showed that mice injected with ESM-1-overexpressing 4T1 and RT-R-4T1 cells exhibited significant increases in tumorigenesis-related molecules (ICAM-1, VCAM-1, MMP-9, HIF-1α, phospho-NF-κB, and phospho-STAT-3) (Figure 6H).

## 3. Discussion

ESM-1 is a secreted proteoglycan dermatan produced by ECs, and as mentioned in the introduction, a high level of ESM-1 secretion has been found in various cancers. In addition, recent reports have suggested that ESM-1 is a biomarker of TNBC [19] and colorectal cancer [31,32]. However, the role of ESM-1 in tumorigenesis in BC, especially RT-R TNBC, is not well known. RT is a main treatment for breast cancer; however, once it acquires radioresistance, it becomes hard to overcome BC. Therefore, in the previous study, we established RT-R-BC cells and studied the changed properties of RT-R-BC cells [5,6]. As mentioned in the introduction, RT-R-MDA-MB-231 cells showed the most radio- and chemo-resistance of tested other RT-R nonTNBC cells and showed increased metastatic properties including increased invasion. In another trial to overcome radioresistance in BC, in this study, we analyzed gene expression levels between MDA-MB-231 cells, a highly metastatic TNBC, and RT-R-MDA-MB-231 cells derived from MDA-MB-231 cells. Interestingly, as shown in Table 1, we found that the ESM-1 was most upregulated in RT-R-MDA-MB-231 cells, compared to MDA-MB-231 cells. Accordingly, we investigated the role of ESM-1 overexpression in the increased tumorigenesis of RT-R-MDA-MB-231 cells. The results of our study showed that at both the mRNA and protein levels, ESM-1 expression was higher in RT-R-MDA-MB-231 cells than in MDA-MB-231 (Figure 1A–C). These results suggest that the overexpression of ESM-1 in RT-R-MDA-MB-231 cells may be the reason why they are more tumorigenic than MDA-MB-231 cells. Actually, in Figure 2A–H, RT-R-MDA-MB-231 cells showed significantly increased proliferation, colony formation ability, migration, and invasion compared to MDA-MB-231 cells, and these effects were significantly reduced by knocking down of ESM-1. A report by Sagara et al. [19] also supports our results; ESM-1 was overexpressed in MDA-MB-231 cells with a brain metastatic phenotype (MDA-MB-231 BR), which are more malignant than MDA-MB-231. The results from Sagara et al. showed that ESM1 gene expression in MDA-MB-231 BR was almost 760-fold higher than that of MDA-MB-231, which is twice as high as our result (320-fold in RT-R-MDA-MB-231 vs. MDA-MB-231), and ESM1 protein level observed in Sagara et al. was also higher than our result. From these results, we could assume that MDA-MB-231 BR might be more aggressive than RT-R-MDA-MB-231. However, the most important common finding is that ESM1 increased in malignant BC cells and play an important role in tumorigenesis of BC.

Compared to normal ECs, BC cells highly express the cell adhesion molecules ICAM-1 and VCAM-1. Our previous study also showed that ICAM-1 and VCAM-1 are involved in tumor growth and migration and the adhesion of cancer cells to other organs [33]. Therefore, in this study, we also examined the role of ESM-1 in cell adhesion molecule expression, the adhesion of cancer cells to ECs, and metastasis-related factors such as MMP-9 and VEGF-A. As expected, compared to MDA-MB-231 cells, RT-R-MDA-MB-231 cells showed increased expression levels of ICAM-1 and VCAM-1 and therefore increased adhesion to ECs, which were significantly inhibited by ESM-1 knockdown (Figure 3A–F). In addition, MMP-9 activity and VEGF-A production were significantly increased in RT-R-MDA-MB-231 cells compared to MDA-MB-231 cells and were significantly reduced in ESM-1 siRNA-transfected RT-R-MDA-MB-231 and MDA-MB-231 cells (Figure 3G,H). These results suggest that the elevated expression of ESM-1 in RT-R-MDA-MB-231 cells may be the reason why RT-R-MDA-MB-231 cells are more tumorigenic than MDA-MB-231 cells.

It has been reported that several signaling molecules affect the tumorigenesis of BC cells. PDK1, which is downstream of phosphoinositide 3-kinase (PI3K) and activated by hormones and growth factors, activates several kinase pathways, including the AKT and PKC pathways, which are involved in the regulation of cell growth, proliferation, survival, and metabolism [34,35]. PDK1 is overexpressed in human BC models, and PDK1 knockdown abolishes tumorigenesis in xenografts [36]. PKCs are activated by the PI3K and PDK1 pathways and then regulate several genes involved in tumor progression, metastasis, and tumorigenesis [37], resulting in the promotion of tumor growth and metastasis [38]. Moreover, PKC increases migration and invasion through the ERK/AP-1 pathway and the production of MMP-9 [39]. The ERK pathway is also involved in tumorigenesis by promoting cell proliferation, survival, and angiogenesis, resulting in decreased life expectancy of TNBC patients [40]. Activation of the ERK pathway potentiates the activation of HIF-1α, NF-κB, and STAT-3, and these transcription factors are well known to be involved in the tumor growth and progression through the transcriptional regulation of various inflammatory genes, such as adhesion molecules, MMPs, and VEGF [26,27,28,29,30,31,32,33,34,35,36,37,38,39,40]. Accordingly, we investigated the effect of ESM-1 on the activation of these intracellular signaling molecules and transcription factors. In our study, RT-R-MDA-MB-231 cells exhibited the increased activation of PDK1, PKC, and ERK1/2 pathways, and moreover, higher expression of HIF-1α and phosphorylation of NF-κB and STAT-3 than MDA-MB-231 cells. However, the expression or the activation of these molecules was reduced in ESM-1 siRNA-transfected RT-R-MDA-MB-231 and MDA-MB-231 (Figure 4 and Figure 5). Interestingly, ESM-1 siRNA-transfected RT-R-MDA-MB-231 cells showed a more dramatic decrease in the expression and activation of these molecules than MDA-MB-231 cells. These results suggest that ESM-1 plays an important role in tumorigenesis in MDA-MB-231 cells and especially RT-R-MDA-MB-231 cells through regulation of these transcription factors.

Finally, we confirmed the role of ESM-1 in tumorigenesis in a mouse xenograft model by subcutaneous injecting normal or ESM-1 expressing 4T1 or RT-R-4T1 murine breast cancer cells into athymic nude mice. In our previous study, mice that were injected with MDA-MB-231 and RT-R-MDA-MB-231 developed tumor mass within a few days but decreased tumor mass as a few days went on, possibly due to the immune barrier. Even though RT-R-MDA-MB-231 increased tumor growth again in the late phase, we failed to observe the successful metastasis to other organs. 4T1 cell is an animal model for stage IV human BC and can mimic the biology of MDA-MB-231, which are highly metastatic BC cells. For this reason, we made a mice xenograft model using 4T1 cells and RT-R-4T1 cells to investigate the role of ESM-1 on tumorigenesis including metastasis in vivo. Furthermore, 4T1 cells do not express ESM-1, so it is good model to study the role of ESM-1 after ESM-1 transfection. Thus, we generated ESM-1-overexpressing 4T1 and RT-R-4T1 cells (Figure 6A). Consistent with the in vitro results, mice injected with ESM-1-overexpressing 4T1 cells exhibited significant increases in tumor volume (Figure 6B–D), lung metastasis (Figure 6F), metastasis-related molecules (VEGF, MMP-9, ICAM-1, and VCAM-1), and transcription factors (HIF-1α, phospho-NF-κB, and phospho-STAT-3) (Figure 6H). This phenomenon was more prominent in the mice injected with ESM-1-overexpressing RT-R-4T1 cells.

## 4. Materials and Methods

### 4.1. Cell Culture

Human BC cell lines, MDA-MB-231, were obtained from the Korea Cell Line Bank (Seoul, Korea), and the mouse BC cell line, 4T1, and human umbilical vascular endothelial cell line, EA.hy 926, were provided by the American Type Culture Collection (ATCC, Manassas, VA, USA). All cancer cell lines were cultured in RPMI-1640, and EA.hy 926 cells were grown in DMEM. Both media were supplemented with 10% fetal bovine serum (FBS), 100 IU/mL penicillin, and 10 μg/mL streptomycin (all from HyClone; GE Healthcare Life Sciences, Logan, UT, USA). The cells were incubated at 37 °C in an incubator containing 5% CO_2_.

### 4.2. Establishment of RT-R-Breast Cancer Cells

RT-R-BC cells (RT-R-MDA-MB-231 cells and RT-R-4T1 cells) were generated from MDA-MB-231 and 4T1 cells as previously described [5]. Briefly, cells were irradiated regularly 25 times at 2 Gy (total 50 Gy), which is a commonly used clinical regimen for the radiotherapy of breast cancer patient, by using a 6-MV photon beam produced by a linear accelerator (Clinac 21EX, Varian Medical Systems, Inc., Palo Alto, CA, USA). After irradiation, cells were grown until they reached approximately 90% confluence, and then, they were trypsinized and subcultured into new flasks. They were irradiated again when the cells reached proximately 70% confluence, which took about 1 week after subculture. The fractionated irradiations were continued until the total dose reached 50 Gy, which totally took about 6 months.

### 4.3. Gene Expression Array Analysis

Total RNA was extracted using TRIzol reagent (Invitrogen, Carlsbad, CA, USA), based on a protocol of the manufacturer, from MDA-MB-231 and RT-R-MDA-MB-231 cells. Then, gene expression profiling was performed by using the QuantiSeq 3′ mRNA-Seq Service (Ebiogen, Seoul, Korea).

### 4.4. Gene Silencing with siRNA

Cells were transfected with 100 nM negative CTRL siRNA or ESM-1 siRNA (Bioneer, Daejeon, Korea) using Turbofect^®^ (Thermo Fisher Scientific, Rockford, IL, USA). The siRNA sequences were as follows: CTRL siRNA forward: 5′-CCUACGCCACCAAUUUCGU-3′; CTRL siRNA reverse: 5′-ACGAA AUUGGUGGCGUAGG-3′; ESM-1 siRNA forward: 5′-CU GAA CAC UUG UAU GUG UU-3′; and ESM-1 siRNA reverse: 5′-AA CAC AUA CAA GYG UUC AG-3′. The cells were incubated in complete medium containing transfection reagent for 24 h, and then the transfection medium was replaced with fresh serum-free medium for starvation. Reverse transcription-polymerase chain reaction (RT-PCR) was performed to determine the efficiency of the gene silencing.

### 4.5. RT-PCR

Extracted total RNA was applied to RT-PCR using TOPscript One-step RT-PCR Drymix (Enzynomics, Daejeon, Korea) according to the manufacturer’s instructions. The sequences of the primers used were as follows: hESM-1 forward: 5′-GC CCT TCC TTG GTA GGT AGC-3′ and reverse: 5′-TG TTT CCT ATG CCC CAG AAC-3′; mESM-1 forward: 5′-ACT CCT GGT ACC TCT GCA CC-3′ and reverse: 5′- CAT TCC ATC CCG AAG GTG CC-3′; hGAPDH forward: 5′- TCA ACA GCG ACA CCC ACT CC-3′ and reverse: 5′-TGA GGT CCA CCC TGT TG-3′; and mGAPDH forward: 5′-GCT GAG TAC GTG GAG-3′ and reverse: 5′-CAT ACT TGG CAG GTT TCT-3′. Thirty cycles of amplification were performed under the following conditions: melting at 95 °C for 30 s, annealing at 57.5 °C or 60 °C for hESM-1 or mESM-1, respectively, for 30 s, and extension at 72 °C for 30 s.

### 4.6. Cell Proliferation Assay

Cell proliferation was analyzed by the CCK-8 assay. CTRL siRNA or ESM-1 siRNA-transfected cells were seeded in 96-well plates (Thermo Fisher Scientific) and incubated for 24 h. Then, cells were added with 10 µL CCK-8 reagent and incubated for an additional 30 min. The optical density of each well was measured at a wavelength of 450 nm using a microplate reader (Tecan, Männedorf, Switzerland).

### 4.7. Colony Formation Assay

Colony formation assays were performed as described previously [6]. Briefly, CTRL siRNA or ESM-1 siRNA-transfected MDA-MB-231 and RT-R-MDA-MB-231 cells (1 × 10^3^ cells/wells) were seeded in 6-well plates and incubated for 10 days. During incubation, the medium was changed every 2–3 days. After 10 days, the medium was discarded, and each well was carefully washed with PBS. The colonies were fixed in 100% methanol for 10 min at room temperature and then stained with 0.1% Giemsa staining solution, the number of visible colonies was counted, and the migrated cells were counted.

### 4.8. Wound Healing Assay

CTRL siRNA or ESM-1 siRNA-transfected MDA-MB-231 and RT-R-MDA-MB-231 cells (2 × 10^5^ cells/well) were cultured in 24-well plates and scratched with a sterile pipette tip. The cells were washed with PBS and incubated at 37 °C in fresh medium for 24 h or 48 h. Images were taken 0, 24, and 48 h after scratching using an Olympus photomicroscope. Cells migrated into scratched area were counted.

### 4.9. Adhesion Assay

ECs were seeded in 6-well plates and cultured until about 90% confluence. CTRL siRNA or ESM-1 siRNA-transfected MDA-MB-231 and RT-R-MDA-MB-231 cells (7.5 × 10^5^ cells/mL, 2 mL/well) were stained with 10 μg/mL of the fluorescent dye BCECF-AM (Boehringer, Mannheim, Germany) at 37 °C for 30 min, and then fluorescence labeled cells were pelleted, resuspended, and added on ECs. After 30 min incubation at 37 °C, the cell suspensions were removed, and the ECs were washed three times with 1 × PBS. The ECs were then visualized under a light microscope (200× magnification) and fluorescent cancer cells were visualized using a fluorescence microscope (Eclipse Ti-U, Nikon, Tokyo, Japan). The number of cancer cells that adhered to ECs was quantified.

### 4.10. Matrigel Invasion Assay

Invasion assays were performed according to Jin et al. [33] with minor modifications, as described below. The upper chambers of inserts were coated with 100 μL of Matrigel (1 mg/mL; BD Biosciences, Franklin Lakes, NJ, USA), and ECs (2 × 10^5^ cells) were added to the Matrigel-coated insert wells. CTRL or ESM-1 siRNA-transfected MDA-MB-231 or RT-R-MDA-MB-231 cells (2 × 10^5^ cells/insert well) in serum-free medium were added to the upper chambers and incubated for 24 h. Then, the noninvaded cells that remained on the upper surface of the insert membranes were removed by scrubbing. The cells that had invaded across the insert well membrane were stained with 4′,6-diamidine-2′-phenylindole dihydrochloride (DAPI, Sigma-Aldrich, St. Louis, MO, USA) and were counted in five randomly selected fields under a fluorescence microscope (Eclipse Ti-U, Nikon).

### 4.11. Gelatin Zymography

Cells (1 × 10^6^ cells) were seeded in 100 mm cell culture dishes and cultured overnight (16 h). Then, the cells were transfected with CTRL or ESM-1 siRNA for 24 h at 37 °C, and the same volume of each conditioned medium was then concentrated 20-fold using protein concentrators (9K MWCO; Thermo Fisher Scientific). The concentrated media were mixed with 2× loading dye and subjected to electrophoresis on 8% sodium dodecyl sulfate-polyacrylamide gel electrophoresis (SDS-PAGE) gels containing 1 mg/mL gelatin. The gels were stained with Coomassie blue solution (0.2% Coomassie brilliant blue R, 50% methanol, and 10% acetic acid) for 30 min and destained with destaining buffer (50% methanol and 10% acetic acid). The enzyme-digested regions that represent MMP-9 activity were identified as white bands on a blue background.

### 4.12. Quantitative VEGF-A Immunoassay

The concentration of VEGF-A in conditioned medium from CTRL or ESM-1 siRNA-transfected MDA-MB-231 and RT-R-MDA-MB-231 cells or the plasma from mice injected with 4T1-empty vector (EV), 4T1-ESM-1, RT-R-4T1-EV, or RT-R-4T1-ESM-1 cells was determined using a VEGF-A enzyme-linked immunosorbent assay kit (R&D Systems, Minneapolis, MN, USA) according to the manufacturer’s instructions.

### 4.13. Quantification of ESM-1 Secretion

Cells (1 × 10^5^ cells) were seeded in 6-well plates and then incubated for 72 h in complete medium at 37 °C. Then, the levels of ESM-1 secreted into cell culture medium were analyzed by using a human ESM-1 quantikine ELISA kit (Novus Biologicals, Centennial, CO, USA) according to the manufacturer’s protocol. All assays were performed in triplicate. The optical density of each well was measured at 450 nm using a microplate reader (Tecan).

### 4.14. Western Blot Analysis

Western blot analysis was performed as described previously [33] with minor modifications. Briefly, aliquots of 30–60 µg of protein were subjected to 8–12% SDS-PAGE and transferred onto Hybond-P^+^ polyvinylidene difluoride membranes (Amersham, Buckinghamshire, UK). The membranes were incubated with the following primary antibodies: anti-ICAM-1 (ab225884, 1:1000, Abcam, Cambridge, UK), anti-VCAM-1 (ab106778, 1:1000, Abcam), anti-phospho-ERK (sc-7383, 1:1000, Santa Cruz Biotechnology), anti-total-ERK (sc-94 1:1000, Santa Cruz Biotechnology), anti-phospho-PKC (9375S, 1:1000, Cell Signaling Technology, Danvers, MA, USA), anti-PKC (sc-10800, 1:1000, Santa Cruz Biotechnology), anti-phospho-PDK1 (3061S, 1:1000, Cell Signaling Technology), anti-PDK1 (3062S, 1:1000, Cell Signaling Technology), anti-phospho-STAT-3 (9131S, 1:1000, Cell Signaling Technology, anti-STAT-3 (4904S, 1:1000, Cell Signaling Technology), anti-HIF-1α (ab2185, 1:1000, Abcam), anti-phospho-NF-κB (8242S, 1:1000, Cell Signaling Technology), anti-NF-κB (8242S 1:1000, Cell Signaling Technology), and anti-β-actin (A2066, 1:2000, Sigma-Aldrich, St. Louis, MO, USA). The bound antibodies were detected with horseradish peroxidase-conjugated secondary antibodies and an ECL western blotting detection reagent (Bio-Rad, Hercules, CA, USA).

### 4.15. ESM-1 Overexpression in 4T1 and RT-R-4T1 cells

4T1 and RT-R-4T1 murine BC were stably transfected with an ESM-1 plasmid vector (PCMV6-kan/Neo, OriGene Technologies, Inc., Rockville, MD, USA), which contained a neomycin resistance gene for the selection of cells stably expressing ESM-1, in serum-free medium using Lipofectamine 3000 (Thermo Fisher Scientific). Following 4 h of incubation at 37 °C, the transfection medium was replaced with fresh medium containing 500 μg/mL neomycin (G-418; Sigma-Aldrich). The culture medium containing neomycin was changed every 2–3 days. Thirty days after transfection, ESM-1 overexpression was confirmed from the total RNA extracted from the cells by RT-PCR.

### 4.16. Animal Experiments

Athymic nude mice (6-week-old females) were purchased from OrientBio (Gyeonggi-do, Korea) and maintained under the following constant ambient conditions: 22 °C to 26 °C; 40% to 60% humidity, 12 h light/dark cycle; and free access to sterilized food and water. Mice were divided into 4 groups (*n* = 7/each group) and injected subcutaneously with 5 × 10^4^ cells/50 μL of 4T1-EV, RT-R-4T1-EV, 4T1-ESM-1, or RT-R-4T1-ESM-1 cells. Body weight and tumor volume were measured twice a week starting from the 7th day after injection. The mice were sacrificed 24th day after injection, and the tumor tissues were fixed in 10% formalin, followed by paraffin infiltration and embedding. Five-micrometer-thick sections were mounted onto ProbeOn Plus microscope slides (Thermo Fisher Scientific), and immunohistochemical analysis was performed using the following primary antibodies; anti-ESM-1 (abx103810, 1:200, Abbexa, Cambridge, UK), ICAM-1 (sc-107, 1:100, Santa Cruz Biotechnology), anti-VCAM-1 (ab106778, 1:100, Abcam), anti-MMP-9 (ab38898, 1:200, Abcam), anti-HIF-1α (ab2185 1:100, Abcam), anti-phospho-NF-κB (3033S, 1:100, Cell Signaling Technology), and anti-phospho-STAT-3 (9145S, 1:100, cell signaling) antibodies. Horseradish peroxidase-conjugated secondary antibodies were used, and then IHC staining was performed using an ABC kit (Vector Labs, Burlingame, CA, USA) and diaminobenzidine (DAB) according to the manufacturer’s instructions. Following DAB staining, the sections were counterstained with Mayer’s hematoxylin solution (Sigma-Aldrich) and observed under a light microscope (CKX41, Olympus). The animal experiment protocol was approved by the Institutional Animal Care and Use Committee at Gyeongsang National University (approval number: GLA-120208-M004, 08 February 2018), and all experiments were performed in compliance with institutional guidelines.

### 4.17. Statistical Analysis

Scanning densitometry was performed using an Image Master^®^ VDS system (Pharmacia Biotech Inc., San Francisco, CA, USA), and intensity ratio of bands were presented in Appendix A. GraphPad Prism 7 software (GraphPad Software, San Diego, CA, USA) was used to analyze all data. One-way ANOVA followed by the Newman-Keuls post hoc test were performed to compare various treatment groups. The data were presented as means ± standard deviation (SD). A *p*-value < 0.05 was considered statistically significant.

## 5. Conclusions

ESM1 is known to be upregulated by inflammatory cytokines (IL-1β, TNF-α, and IFN-γ) and proangiogenic factors (VEGF-A and VEGF-C) and PI3K dependent pathway in cancer cell or endothelial cell [9,10,11,12]. However, the intracellular mechanisms induced by ESM1 are not well known. Kang et al. [17] reported that ESM1 plays a role in cell survival, cell cycle, migration, and invasion in colorectal cancer through affecting NF-κB and phospho-Akt pathways. Our results suggest for the first time that ESM-1 is the most overexpressed gene in RT-R-MDA-MB-231 cells compared to MDA-MB-231 cells and plays a critical role in tumorigenesis in breast cancer (Figure 7) through regulation of PDK, PKC, and ERK1/2 pathways and the subsequent activation of transcription factors HIF-1α, NF-κB, and STAT-3 to regulate adhesion molecules, MMPs, and VEGF. Based on these findings, ESM-1 may be a target molecule for treating BC, especially TNBC.

## Figures and Tables

**Figure 1 cancers-12-01363-f001:**
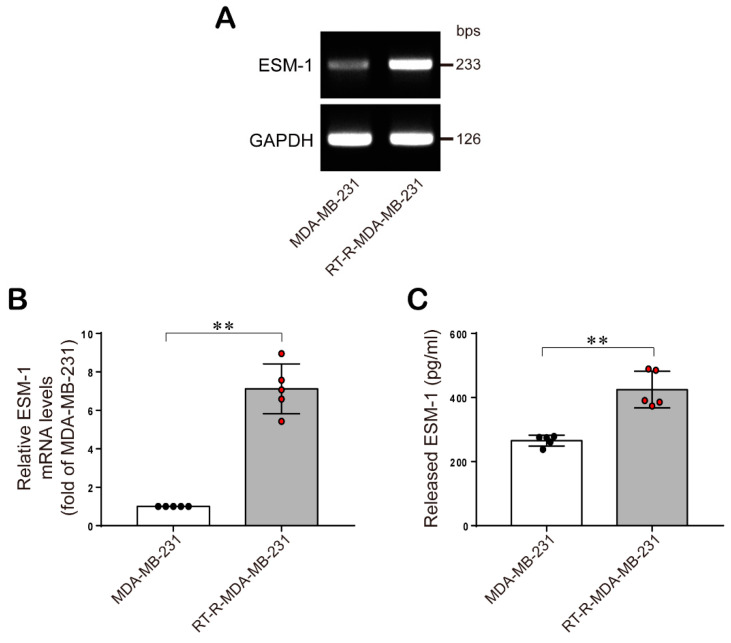
ESM-1 expression levels in MDA-MB-231 cells and their RT-R derivatives. (**A**,**B**) ESM-1 mRNA expression in RT-R-MDA-MB-231 cells and their parental cells, MDA-MB-231 was analyzed by RT-PCR. The full blot image can be found in Appendix A. Representative ESM-1 mRNA levels (**A**) and the quantification of ESM-1 mRNA levels (**B**) in MDA-MB-231 and RT-R-MDA-MB-231 cells. The data represent the mean ± SD of five independent experiments. (**C**) Cells were incubated for 72 h, and the levels of ESM-1 secreted by MDA-MB-231 and RT-R-MDA-MB-231 cells into cell culture medium were measured using ELISA kit for ESM-1 as described in the methods sections. The data represent the mean ± SD of five independent experiments. * *p* < 0.05, ** *p* < 0.01.

**Figure 2 cancers-12-01363-f002:**
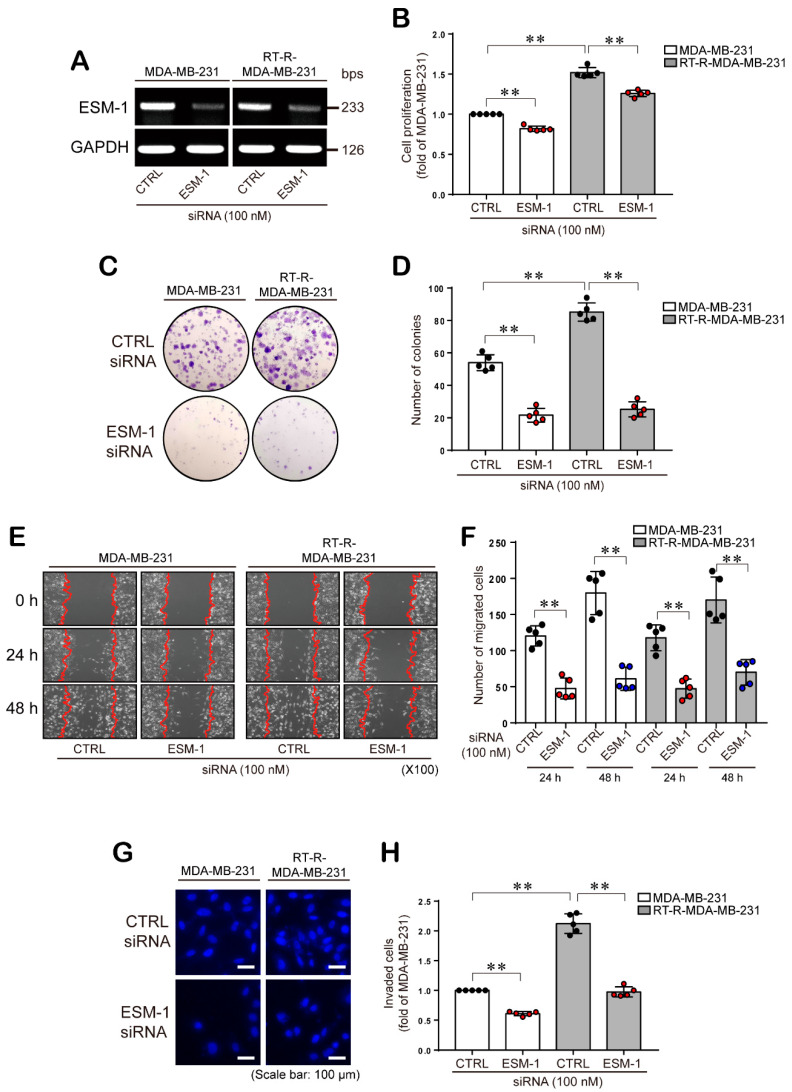
ESM-1 plays an important role in the proliferation, colony formation ability, migration, and invasion of both MDA-MB-231 and RT-R- MDA-MB-231 cells. (**A**) Cells were transfected with CTRL or ESM siRNA (100 nM) as described in the methods section, and gene silencing efficiency in MDA-MB-231 and RT-R-MDA-MB-231 cells was determined by RT-PCR. The full blot image can be found in Appendix A. (**B**) CTRL or ESM-1 siRNA-transfected MDA-MB-231 and RT-R-MDA-MB-231 cells were subjected to a cell proliferation assay using CCK-8 reagent as described in the methods section. The data represent the mean ± SD of five independent experiments. (**C**,**D**) CTRL or ESM-1 siRNA-transfected cells (1 × 10^3^) were seeded in 6-well plates, and 10 days later, colonies were stained with Giemsa solution (0.1%) (**C**) and quantified under a light microscope as described in the methods section (**D**). The data represent the mean ± SD of five independent experiments. (**E**) CTRL or ESM-1 siRNA-transfected cells were seeded in 6-well plates, and a wound scratch assay was performed as described in the methods section. (**F**) The migrated cells were quantified under a microscope 0, 24, and 48 h after wound scratching. The data represent the mean ± SD of five independent experiments. (**G**,**H**) CTRL or ESM-1 siRNA-transfected cells were added to EC-Matrigel-coated insert wells and incubated for 16 h at 37 °C. The invaded cells that had invaded across the membrane were stained with DAPI (**G**), and five randomly selected fields were counted under a fluorescence microscope (**H**). The data represent the mean ± SD of five independent experiments. ** *p* < 0.01.

**Figure 3 cancers-12-01363-f003:**
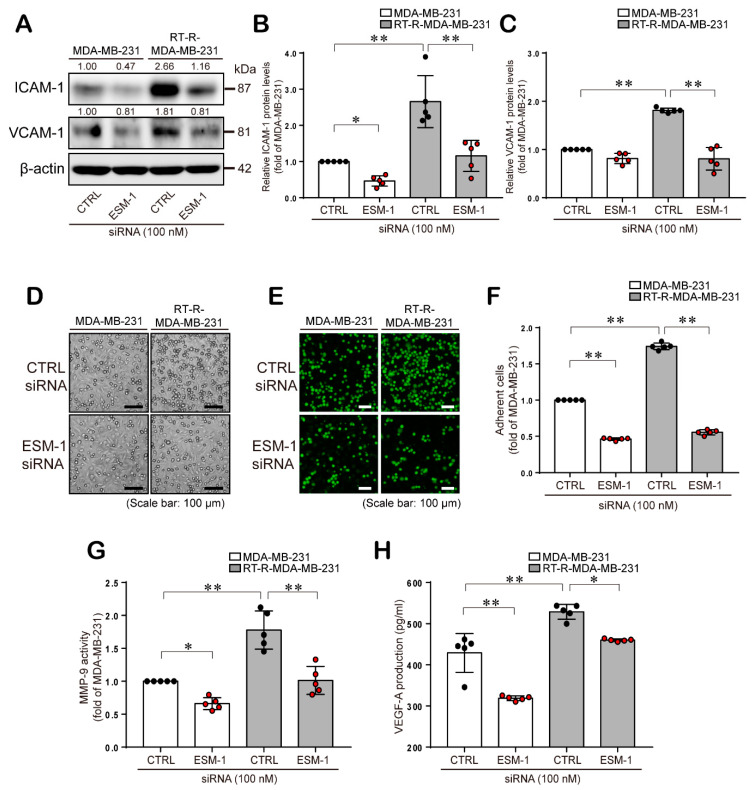
ESM-1 knockdown reduces ICAM-1 and VCAM-1 protein levels, resulting in decreases in the adhesion of cancer cells to ECs, MMP-9 activity, and VEGF-A production in MDA-MB-231 and RT-R-MDA-MB-231 cells. (**A**) ICAM-1 and VCAM-1 protein levels were analyzed in CTRL or ESM-1 siRNA-transfected MDA-MB-231 and RT-R-MDA-MB-231 cells by western blotting. The full blot image can be found in Appendix A. (**B**,**C**) Relative protein levels of ICAM-1 (**B**) and VC. AM-1 (**C**) were quantified. The data represent the mean ± SD of five independent experiments. (**D**–**F**) CTRL or ESM-1 siRNA-transfected cells were added to ECs for 30 min, and representative images of the adhesion of cancer cells to ECs are shown using a light microscope (**D**) and a fluorescence microscope (**E**). BC cells that had adhered to ECs were quantified in five randomly selected fields under a fluorescence microscope (**E**). (**G**,**H**) Cells were transfected with CTRL or ESM-1 siRNA, and MMP-9 activity (**G**) and VEGF-A production (**H**) were detected in conditioned media by zymography and ELISA, respectively. The data represent the mean ± SD of five independent experiments. * *p* < 0.05, ** *p* < 0.01.

**Figure 4 cancers-12-01363-f004:**
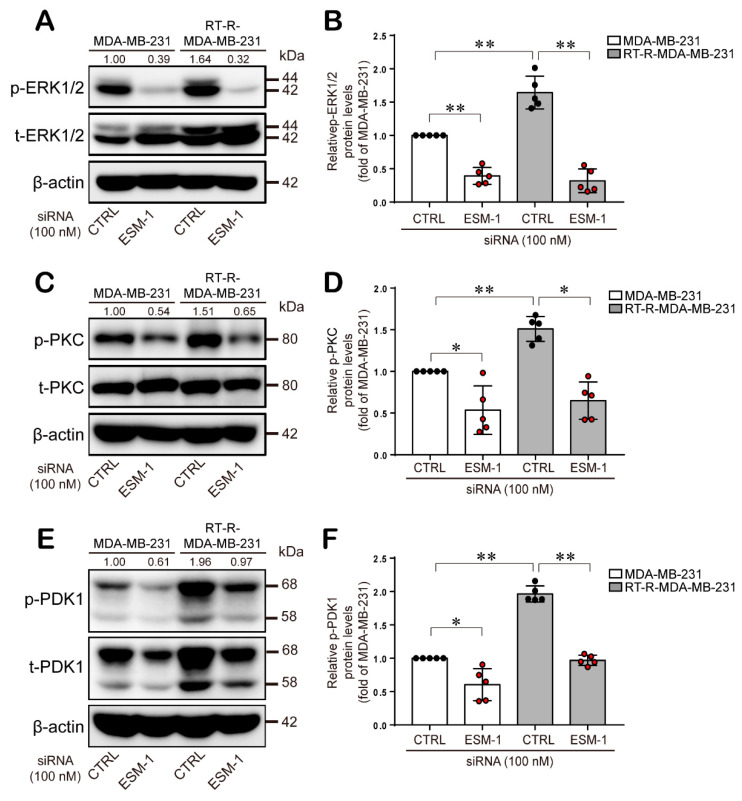
ERK1/2, PKC, and PDK1 phosphorylation are enhanced in RT-R-MDA-MB-231 cells compared to MDA-MB-231 cells and are suppressed by knockdown of ESM-1. Cell lysates were obtained from the CTRL- or ESM-1 siRNA-transfected cells, and western blotting was performed to detect the expressions of phospho-ERK1/2/ERK1/2 (**A**), phospho-PKC/PKC (**C**), and phospho-PDK1/PDK1 (**E**). Band densities were quantified, and the relative protein levels are presented as the mean ± SD of five independent experiments (**B**,**D**,**F**). * *p* < 0.05, ** *p* < 0.01. The full blot images can be found in Appendix A.

**Figure 5 cancers-12-01363-f005:**
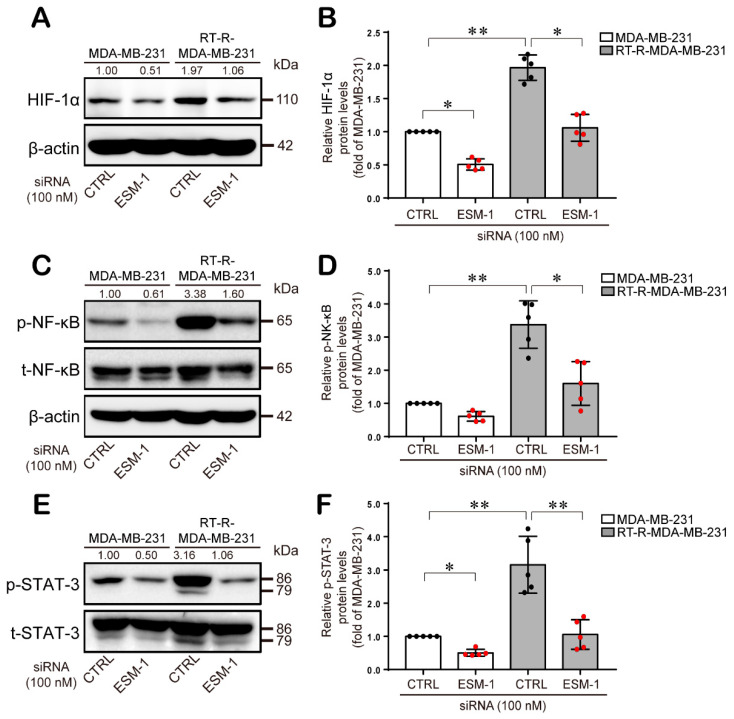
HIF-1α induction and NF-κB and STAT-3 phosphorylation are increased in RT-R-MDA-MB-231 cells compared to MDA-MB-231 cells and are downregulated by ESM-1 knockdown. Cell lysates were obtained from the CTRL- or ESM-1 siRNA-transfected cells, and western blotting was performed to detect HIF-1α (**A**), phospho-NF-κB/NF-κB (**C**), and phospho-STAT-3/STAT-3 expression (**E**). Band densities were quantified, and the relative protein levels are presented as the mean ± SD of five independent experiments (**B**,**D**,**F**). * *p* < 0.05, ** *p* < 0.01. The full blot images can be found in Appendix A.

**Figure 6 cancers-12-01363-f006:**
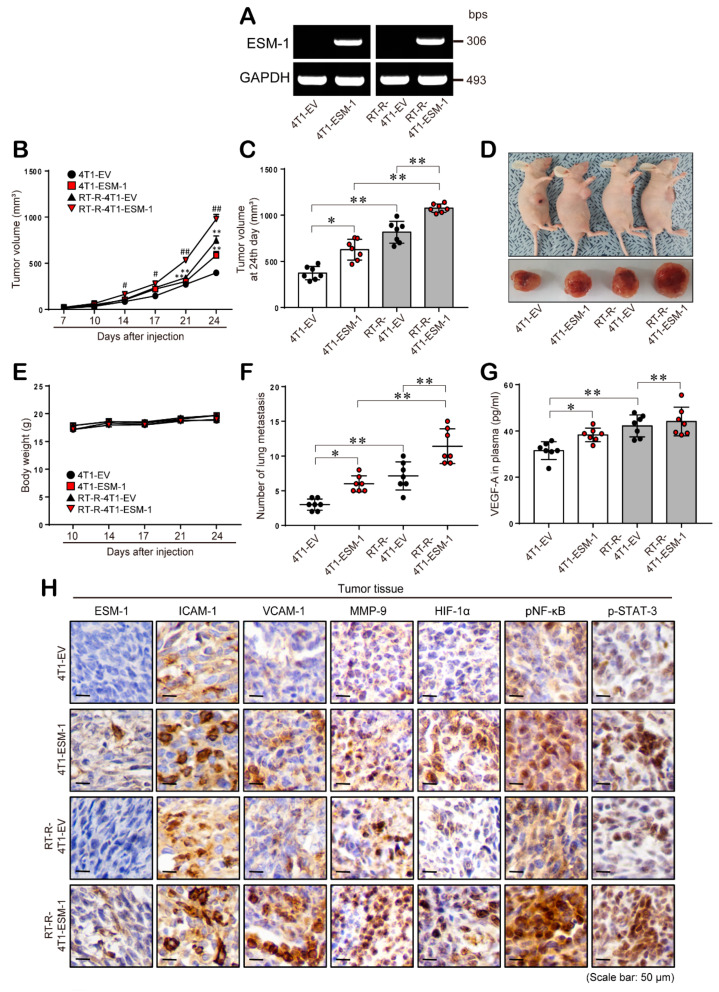
ESM-1 overexpression increases tumor growth and metastasis in an in vivo mouse model. (**A**) 4T1 and RT-R-4T1 mouse breast cancer (BC) cells were stably transfected with expression plasmid vectors encoding ESM-1 (4T1-ESM-1 and RT-R-4T1-ESM-1 cells) or with an empty vector (4T1-EV and RT-R-4T1-EV cells) as described in the methods section. The full blot image can be found in Appendix A. (**B**–**H**) Athymic nude mice were divided into four groups and injected subcutaneously with 4T1-EV cells (4T1; *n* = 7), RT-R-4T1-EV cells (*n* = 7), 4T1-ESM-1 cells (*n* = 7), or RT-R-4T1-ESM-1 cells (*n* = 7) (5 × 10^4^ cells/50 μL of serum-free medium). The mice were sacrificed on the 24th day, and the tumors and lung tissues were extracted. Tumor volume (**B**–**D**) and body weight (**E**) were measured every three days during tumor development. The incidence of lung metastasis (**F**) was examined after sacrifice. (**G**) VEGF-A was detected in the plasma by ELISA. The data represent the mean ± SD. * *p* < 0.05, ** *p* < 0.01. (**H**) Tumor tissue sections were stained with anti-ESM-1, ICAM-1, VCAM-1, MMP-9, HIF-1α, phospho-NF-κB, and phospho-STAT-3 antibodies.

**Figure 7 cancers-12-01363-f007:**
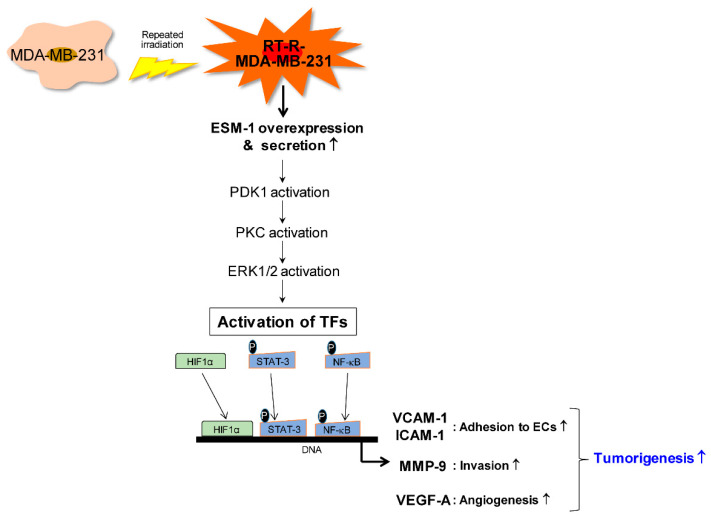
Schematic presentation of the mechanism by which ESM-1 plays an important role in the tumorigenesis of breast cancer cells.

**Table 1 cancers-12-01363-t001:** Top 10 upregulated genes in RT-R-MDA-MB-231 compared to MDA-MB-231.

Gene Symbol	Fold Change (RT-R-MDA-MB-231 vs. MDA-MB-231)
ESM1	318.75
MMP1	203.77
SEL1L3	131.56
MAGEB2	118.04
ALPK2	116.37
LOC100134317	100.11
PHGDH	98.82
LOC284412	97.10
PLCB4	89.00
PLCH2	82.37

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
