# Peer review of "ESM-1 Overexpression is Involved in Increased Tumorigenesis of Radiotherapy-Resistant Breast Cancer Cells"

_cancers, 2020, doi:10.3390/cancers12061363_

Round 1

Reviewer 1 Report

Comments to Jin et al.

This is an original and well-written paper, in line with the journal’s aim and scope, where the main objective is to describe ESM-1 overexpression as involved in increased tumorigenesis of radiotherapy-resistant breast cancer cells.

The research was executed according to currently accepted standards, approach to statistical analysis of data is appropriate and the manuscript provides data that are likely to be of interest to journal readers. Summarizing, the paper is interesting to read, however, some corrections should be applied and listed below:

  1. Please, use acronyms for sentences (i.e. Breast cancer - BC, radiotherapy -RT), often used in the manuscript. The first time the acronym is used, it should be fully written out.
  2. The authors should better organize abstract structure, the aim is not clearly highlighted, results and conclusion were often mixed.
  3. In order to understand the intracellular mechanisms induced by ESM1, the authors should report some literature data regarding intracellular networks in which this protein is included as a key regulator (also by using a figure). In this sense, it's up and down effectors could be also highlighted and assayed.
  4. Regarding the reference section, 18 out of 39 bibliographies were published before 2010. the authors should provide more updated bibliographic data.
  5. Please include some information regarding RT schedules often used in breast cancer care with a special focus in fractionated (in term of ionizing radiation doses) RT plans.
  6. In addition, the authors often use their previous results to introduce their hypothesis on this work. In this sense, the reader has to read the author's previous work (in some part in the same time), to better understand the topics highlighted in this paper. Although this kind of approach is correctly used in material and methods section, I suggest introducing some precious information included in reference 5 previously published by the authors, in order to facilitate reader comprehension (for example in the introduction section).
  7. Please clarify the time window used during fractionated irradiation to select radioresistant MDA-MB-231 cells. In turn, the authors in their previous work (and not in this paper), have written that the fractionated irradiations were continued until the total dose reached 50 Gy, but no information regarding the time windows used to obtain this total dose is available.
  8. As described in the text, radioresistant MDA-MB-231 cells up-regulate some genes involved in epithelial-mesenchymal transition (EMT). The activation of this key process involved in cell aggressiveness and migration was assayed by the authors?
  9. Please, add values to all western blotting figures in order to facilitate protein trend variation comprehension.
  10. As described in a previous work of the same authors (reference n.5), among three breast cancer cell lines used, only MDA‑MB‑231 cells derived from highly metastatic MDA‑MB‑231 cells exhibited most radioresistance. Thus, I consider redundant the description of MCF7 and T47D gene expression results included in figure 1 because these cell lines were not used in the other parts of this paper.
  11. Finally, considering that all the paper described the gene expression trend of MDA-MB-231, RT-R-MDA‑MB‑231 (with and without ESM1 knocked down), the author must explain why mice were not xenotransplanted with MDA-MB-231 cell line. As known, 4T1 mouse cell line is an animal model for stage IV human BC even if is often used as a post-operative model as well as a non-surgical model because the 4T1-induced tumor metastasizes spontaneously in both models with similar kinetics. As known, BC is a heterogeneous disease classified in several subgroups based on molecular and genomic profiles, associated with different treatment response. Among these, the triple-negative breast cancer (TNBC) subtype (and MDA-MB-231 cell line is an in vitro model of this subgroup), is characterized by unsuccessful therapies due to the absence of hormonal or targeted therapy against it, makes it a clinical challenge for oncologists in terms of patient management. Anyhow, the authors described interesting results also by using 4T1 mouse even if, in my opinion, does not represent the right xenograft model that should be used by the authors in order to have more robust data on topic investigated (TNBC cells).
  12. Please, correct the reference style in the text (for example reference 17), according to journal standard.
  13. How many mice were included in each experimental group?
  14. If available, please provide the protocol number of the approved project by Ethic committee.

Author Response

Comments and Suggestions for Authors

This is an original and well-written paper, in line with the journal’s aim and scope, where the main objective is to describe ESM-1 overexpression as involved in increased tumorigenesis of radiotherapy-resistant breast cancer cells.

The research was executed according to currently accepted standards, approach to statistical analysis of data is appropriate and the manuscript provides data that are likely to be of interest to journal readers. Summarizing, the paper is interesting to read, however, some corrections should be applied and listed below:

  1. Please, use acronyms for sentences (i.e. Breast cancer - BC, radiotherapy -RT), often used in the manuscript. The first time the acronym is used, it should be fully written out.

→ Answer: We used the abbreviation BC and RT for breast cancer and radiotherapy, respectively. In addition, we added more abbreviations after definition. Please see the revised manuscript.

  1. The authors should better organize abstract structure, the aim is not clearly highlighted, results and conclusion were often mixed.

→ Answer: Thank you for your comment. We have reviewed and reorganized Abstract; tried to make clear the aim. Please see the revised Abstract (lines 15~19 in red).

  1. In order to understand the intracellular mechanisms induced by ESM1, the authors should report some literature data regarding intracellular networks in which this protein is included as a key regulator (also by using a figure). In this sense, it's up and down effectors could be also highlighted and assayed.

→ Answer: Thank you for your comments.

ESM1 is known to be upregulated by inflammatory cytokines (IL-1b, TNF-a, IFN-g) and proangiogenic factors (VEGF-A and VEGF-C) and PI3K dependent pathway in cancer cell or endothelial cell (Bechard et al, 2000; Lassalle et al., 1996; Rennel et al., 2007; Shin et al., 2008). However, there is very few report about the intracellular mechanisms induced by ESM1; Kang et al (2012) reported that ESM1 plays a role in cell survival, cell cycle, migration and invasion in colorectal cancer through affecting NF-kB and phospho-Akt pathway.

As you suggested, in order to understand the intracellular mechanisms induced by ESM1, we searched literatures and found that several signaling molecules such as PDK1, PKC and ERK1/2 pathways affect the tumorigenesis of BC cells. These pathways are related to transcription factors HIF-1α, NF-kB and STAT-3 which are involved in the tumor growth and progression through the transcriptional regulation of various inflammatory genes, such as adhesion molecules, MMPs, and VEGF. Therefore, in this study, we tried to investigate whether ESM-1 overexpressed in RT-R-MDA-MB-231 cells affects PDK1, PKC and ERK1/2 pathways and the transcription factors HIF-1α, NF-kB and STAT-3 to regulate adhesion molecules, MMPs, and VEGF.

Regarding your questions, we have already described these points in the Discussion as follows (lines 280~293); “It has been reported that several signaling molecules affect the tumorigenesis of BC cells. PDK1, which is downstream of phosphoinositide 3-kinase (PI3K) and activated by hormones and growth factors, activates several kinase pathways, including the AKT and PKC pathways, which are involved in the regulation of cell growth, proliferation, survival and metabolism [29,30]. PDK1 is overexpressed in human BC models, and PDK1 knockdown abolishes tumorigenesis in xenografts [31]. PKCs are activated by the PI3K and PDK1 pathways and then regulate several genes involved in tumor progression, metastasis and tumorigenesis [32], resulting in the promotion of tumor growth and metastasis [33]. Moreover, PKC increases migration and invasion through the ERK/AP-1 pathway and the production of MMP-9 [34]. The ERK pathway is also involved in tumorigenesis by promoting cell proliferation, survival and angiogenesis, resulting in decreased life expectancy of TNBC patients [35]. Activation of the ERK pathway potentiates the activation of HIF-1α, NF-kB and STAT-3, and these transcription factors are well known to be involved in the tumor growth and progression through the transcriptional regulation of various inflammatory genes, such as adhesion molecules, MMPs, and VEGF [36-40]”.

During revision, according to your suggestion, we investigated whether ESM-1 overexpressed in RT-R-MDA-MB-231 cells affects PDK1, PKC and ERK1/2 pathways as key regulators and added some results in Figure 4 (Please see new Figure 4; lines 293~298). In addition, we modified Figure 7 and included these key regulators in the schematic mechanism by which ESM-1 plays an important role in the tumorigenesis of breast cancer cells. Furthermore, we changed Conclusion part including these points. Please see new Figure 7 and Conclusions (lines 471~ 480).

Bechard D, Meignin V, Scherpereel A, Oudin S, Kervoaze G, Bertheau P, Janin A, Tonnel A, Lassalle P. Characterization of the secreted form of endothelial-cell-specific molecule 1 by specific monoclonal antibodies. J Vasc Res. 2000 Sep-Oct;37(5):417-25.

Lassalle P, Molet S, Janin A, Heyden JV, Tavernier J, Fiers W, Devos R, Tonnel AB. ESM-1 is a novel human endothelial cell-specific molecule expressed in lung and regulated by cytokines. J Biol Chem. 1996 Aug 23;271(34):20458-64.

Rennel E, Mellberg S, Dimberg A, Petersson L, Botling J, Ameur A, Westholm JO, Komorowski J, Lassalle P, Cross MJ, Gerwins P. Endocan is a VEGF-A and PI3K regulated gene with increased expression in human renal cancer. Exp Cell Res. 2007 Apr 15;313(7):1285-1294.

Shin JW, Huggenberger R, Detmar M. Transcriptional profiling of VEGF-A and VEGF-C target genes in lymphatic endothelium reveals endothelial-specific molecule-1 as a novel mediator of lymphangiogenesis. Blood. 2008 Sep 15;112(6):2318-26.

Kang YH, Ji NY, Han SR, Lee CI, Kim JW, Yeom YI, Kim YH, Chun HK, Kim JW, Chung JW, Ahn DK, Lee HG, Song EY. ESM-1 regulates cell growth and metastatic process through activation of NF-κB in colorectal cancer. Cell Signal. 2012 Oct;24(10):1940-9.

  1. Regarding the reference section, 18 out of 39 bibliographies were published before 2010. the authors should provide more updated bibliographic data.

→ Answer: According to your suggestion, we tried to provide more updated references and replaced some of them with new ones which had similar contents with the previous ones (please see the reference list in red). However, in some cases, we didn't change references because we couldn’t find the similar ones with the previous references, or the previous one was first report which had originality. Please understand this point.

  1. Please include some information regarding RT schedules often used in breast cancer care with a special focus in fractionated (in term of ionizing radiation doses) RT plans.

→ Answer: According to the comment of Radiation oncologists, total 50 Gy radiation in 2 Gy fractions is a commonly used clinical regimen for the radiotherapy of breast cancer patient. In addition, it is reported that standard radiotherapy protocol involves daily exposures of 2 Gy fraction dose for few weeks, where patients receive a cumulative dose of 50 Gy to 70 Gy during radiotherapy course (Yasser et al., 2017; Wang et al., 2005). We added this information briefly in the lines 331.

In addition, we included some information regarding RT schedules which were used to make RT-R-BC in the lines 333~336 as follows;

“After irradiation, cells were grown until they reached approximately 90% confluence, and then, they were trypsinized and subcultured into new flasks. They were irradiated again when the cells reached proximately 70% confluence, which took about 1 week after subculture. The fractionated irradiations were continued until the total dose reached 50 Gy, which totally took about 6 months”.

Yasser M, Shaikh R, Chilakapati MK, Teni T. Raman spectroscopic study of radioresistant oral cancer sublines established by fractionated ionizing radiation. PLoS One. 2014, 9(5):e97777.

Wang T, Tamae D, LeBon T, Shively JE, Yen Y, Li JJ. The role of peroxiredoxin II in radiation-resistant MCF-7 breast cancer cells. Cancer Res. 2005, 65:10338–46.

  1. In addition, the authors often use their previous results to introduce their hypothesis on this work. In this sense, the reader has to read the author's previous work (in some part in the same time), to better understand the topics highlighted in this paper. Although this kind of approach is correctly used in material and methods section, I suggest introducing some precious information included in reference 5 previously published by the authors, in order to facilitate reader comprehension (for example in the introduction section).

→ Answer: Thank you for your comments. We totally agree with your opinion. So, we rewrote the Introduction part (highlighted in red: lines 53~70) to make the readers better understand the background and the aim of our present study.

  1. Please clarify the time window used during fractionated irradiation to select radioresistant MDA-MB-231 cells. In turn, the authors in their previous work (and not in this paper), have written that the fractionated irradiations were continued until the total dose reached 50 Gy, but no information regarding the time windows used to obtain this total dose is available.

→ Answer: We have already answered in Question # 5. Please see above.

  1. As described in the text, radioresistant MDA-MB-231 cells up-regulate some genes involved in epithelial-mesenchymal transition (EMT). The activation of this key process involved in cell aggressiveness and migration was assayed by the authors?

→ Answer: In previous study, we determined that RT-R-MDA-MB-231 cells exhibits more invasive property than their parental MDA-MB-231 cells through upregulation of EMT-related molecules such as Snail 1 and β-catenin as well as increase of MMP-9 activity (5), and we described this point in the Introduction (lines 61, 62). Although we did not use a specific inhibitor or targeted siRNA of the molecules in cell aggressiveness and migration in RT-R-MDA-MB-231 cells, our previous study (Ko et al., 2018) showed that oleandrin and odoroside A, polyphenolic monomer compounds which suppressed β-catenin expression level and MMP-9 activity, were able to inhibit invasion in RT-R-MDA-MB-231 and MDA-MB-231 cells.

Ko YS, Jin H, Lee JS, Park SW, Chang KC, Kang KM, Jeong BK, Kim HJ. Radioresistant breast cancer cells exhibit increased resistance to chemotherapy and enhanced invasive properties due to cancer stem cells. Oncol Rep. 2018, 40(6):3752-3762.

Ko YS, Rugira T, Jin H, Park SW, Kim HJ. Oleandrin and Its Derivative Odoroside A, Both Cardiac Glycosides, Exhibit Anticancer Effects by Inhibiting Invasion via Suppressing the STAT-3 Signaling Pathway. Int J Mol Sci. 2018B, 19(11). pii: E3350.

  1. Please, add values to all western blotting figures in order to facilitate protein trend variation comprehension.

→ Answer: We added the values above each blots. Please see the updated figures (Fig. 3A; Fig. 4A, C, E; Fig. 5A, C, E).

  1. As described in a previous work of the same authors (reference n.5), among three breast cancer cell lines used, only MDA‑MB‑231 cells derived from highly metastatic MDA‑MB‑231 cells exhibited most radioresistance. Thus, I consider redundant the description of MCF7 and T47D gene expression results included in figure 1 because these cell lines were not used in the other parts of this paper.

→ Answer: Thank you for your valuable comment. According to your suggestion, we deleted the mRNA and protein data for MCF7 and T47D from Figure 1. Accordingly, we modified the results section and Discussion section. Please see new Figure 1, Results (lines 85~91) and Discussion (lines 245~249; 252~255, in red).

  1. Finally, considering that all the paper described the gene expression trend of MDA-MB-231, RT-R-MDA‑MB‑231 (with and without ESM1 knocked down), the author must explain why mice were not xenotransplanted with MDA-MB-231 cell line. As known, 4T1 mouse cell line is an animal model for stage IV human BC even if is often used as a post-operative model as well as a non-surgical model because the 4T1-induced tumor metastasizes spontaneously in both models with similar kinetics. As known, BC is a heterogeneous disease classified in several subgroups based on molecular and genomic profiles, associated with different treatment response. Among these, the triple-negative breast cancer (TNBC) subtype (and MDA-MB-231 cell line is an in vitro model of this subgroup), is characterized by unsuccessful therapies due to the absence of hormonal or targeted therapy against it, makes it a clinical challenge for oncologists in terms of patient management. Anyhow, the authors described interesting results also by using 4T1 mouse even if, in my opinion, does not represent the right xenograft model that should be used by the authors in order to have more robust data on topic investigated (TNBC cells).

→ Answer: We appreciate your valuable comments. In our previous study, we did in vivo experiments using mice which were injected with MDA-MB-231 and RT-R-MDA-MB-231. When we injected MDA-MB-231 and RT-R-MDA-MB-231 to the mice, mice developed tumor mass within a few days but decreased tumor mass as a few days went on, because human BC cells couldn’t overcome immune barrier of mice, we think. Even though RT-R-MDA-MB-231 increased tumor growth again in the late phase, we failed to observe the successful metastasis to other organ.

According to other researchers including Ferrari-Amorotti G et al. performed BC study using both cell lines human BC cells such as MDA-MB-231 and mouse BC cells such as 4T1 cell (Ferrari-Amorotti G et al., 2014; Larive RM et al., 2014; Zhou R et al., 2014). 4T1 cell is an animal model for stage IV human BC as you mentioned, so we expected that it can mimic the biology of MDA-MB-231 which is highly metastatic BC cells. To investigate the role of ESM-1 on tumorigenesis including metastasis in vivo, we made mice xenograft model using 4T1 and RT-R-4T1. Furthermore, 4T1 cell doesn’t express ESM-1, so it is good model to study the role of ESM-1 after ESM-1 transfection, we think. In our experiments, we stably transfected ESM-1 into 4T1 and RT-R-4T1 cells as described in the Methods (4.15 ESM-1 overexpression in 4T1 and RT‑R‑4T1 cells), and injected to mice.

To help readers’ understanding, we explained this point briefly in the Discussion (please see lines 305~312).

Ferrari-Amorotti G, Chiodoni C, Shen F, Cattelani S, Soliera AR, Manzotti G, Grisendi G, Dominici M, Rivasi F, Colombo MP, Fatatis A, Calabretta B. Suppression of invasion and metastasis of triple-negative breast cancer lines by pharmacological or genetic inhibition of slug activity. Neoplasia. 2014, 16(12):1047-58.

Larive RM, Moriggi G, Menacho-Márquez M, Cañamero M, de Álava E, Alarcón B, Dosil M, Bustelo XR. Contribution of the R-Ras2 GTP-binding protein to primary breast tumorigenesis and late-stage metastatic disease. Nat Commun. 2014, 5:3881.

Zhou R, Xu L, Ye M, Liao M, Du H, Chen H. Formononetin inhibits migration and invasion of MDA-MB-231 and 4T1 breast cancer cells by suppressing MMP-2 and MMP-9 through PI3K/AKT signaling pathways. Horm Metab Res. 2014, 46(11):753-60.

  1. Please, correct the reference style in the text (for example reference 17), according to journal standard.

→ Answer: We corrected them according to standard.

  1. How many mice were included in each experimental group?

→ Answer: Figure 6 legend already includes the information about mice number used in in vivo experiments (n = 7) (lines 231, 232). To help the readers’ understanding, we also included that information in the Method section. Please see lines 448.

  1. If available, please provide the protocol number of the approved project by Ethic committee.

→ Answer: It has already included in the lines 461~464 as follows; “The animal experiment protocol was approved by the Institutional Animal Care and Use Committee at Gyeongsang National University (approval number: GLA-120208‑M004), and all experiments were performed in compliance with institutional guidelines”.

Reviewer 2 Report

Dear authors, thank you for a very intersting paper.

Please find my comments and suggestions also in the attached file.

  1. paragraph is written very confusingly. Especially, having a look into the results, RT cells of MCF7 and T47D do show a decreased expression of ESM-1. This is not sufficiently discussed. (line 82-89)
  2. page 6 Figure 2 C quality has to be improved
  3. pag 8 Figure 3 D and Methods part: How did the authors count the adhering breast cancer cells? In addition, it seems that there is too much uncovered petri dish to determine if the breast cancer cells realy stick to the ECs or the petri dish. This experiment needs to be repeated using confluent ECs and dicrimination markers of ECs and MDA-MB-231 to give trustworthy results.
  4. page 12 lines 234-235: The results of MCF7 and 4T1 cells should also be discussed.
  5. line 275 seems to be wrong style
  6. page 14 wound healing: How did the authors analyse the images?
  7. page 14 adhesion assay: That seems to be a rather subjective method. The images do not give sufficient space for analysis. It also seems that ECs are not at 90% confluence. This analysis should be repeated using factors which enable discrimination between ECs and breast cancer cells to give trustworthy results.
  8. page 14 matrigel invasion assay: how did the authors count the cells?
  9. page 15 Gelatin zymography: If you use supernatent you have to be sure about the number of cells which had been seeded in the first place. How many cells have been seeded prior to the experiment?
  10. In general: how many biological replicates have been used?
  11. page 15 pont 4.15: Were the cells stably transfected or transient? Had transfected clones been selected or did the authors used a hetergenous culture of undefined transfected cells? It is a huge difference if only one clone is used or different clones in one culture.
  12. page 16 point 4.16: Using standard error of the mean always gives the "herb taste" that the results graphs had to be polished. SD would be a better choice to show visible representative data.
  13. page 16 Conclusion: please enhance image quality

Author Response

Comments and Suggestions for Authors

Dear authors, thank you for a very interesting paper.

Please find my comments and suggestions also in the attached file.

  1. paragraph is written very confusingly. Especially, having a look into the results, RT cells of MCF7 and T47D do show a decreased expression of ESM-1. This is not sufficiently discussed. (line 82-89)

→ Answer: Thank you for your valuable comment. Reviewer 1 also gave a similar comment; he(she) suggested to delete the mRNA and protein data for MCF7 and T47D from Figure 1. Because, in our previous study, among three breast cancer cell lines used, only MDA‑MB‑231 cells derived from highly metastatic MDA‑MB‑231 cells exhibited most radioresistance, and MCF7 and T47D cell lines were not used in the other parts of this paper. Thus, we deleted the data for MCF7 and T47D from Figure 1. Instead, we inserted some sentence discussing about the data of MCF-7 and T47D data in Discussion part. Please see new Figure 1, Results (lines 85~91) and Discussion (lines 245~249; 252~255 in red).

  1. page 6 Figure 2 C quality has to be improved.

→ Answer: As you suggested, we improved quality of the figure 2C. Please see Figure 2C. Compared to other cancer cells, MDA-MB-231 cells tend to form colonies in a dispersed form under our experimental protocol. When we refer to other protocols, the soft agar methods (anchorage-independent growth capacity assay) might be useful for these cells. So, now, we are trying to establish a new protocol using soft agar. Please understand this point.

  1. page 8 Figure 3 D and Methods part: How did the authors count the adhering breast cancer cells? In addition, it seems that there is too much uncovered petri dish to determine if the breast cancer cells really stick to the ECs or the petri dish. This experiment needs to be repeated using confluent ECs and discrimination markers of ECs and MDA-MB-231 to give trustworthy results.

→ Answer: According to your comments, we changed the experimental Methods and changed the Result. Please see 4.9. Adhesion assay (lines 381~388) and new Figure 3D and 3E. Briefly, cancer cells were stained with fluorescent dye BCECF-AM and then added on ECs. ECs were visualized under a light microscope (Fig. 3D) and fluorescent cancer cells were visualized using the Eclipse Ti-U inverted microscope (Nikon, Tokyo, Japan) (Fig. 3E). The number of cancer cells that adhered to ECs was quantified (Fig. 3F).

  1. page 12 lines 234-235: The results of MCF7 and 4T1 cells should also be discussed.

→ Answer: As we answered above, we deleted the data for MCF7 and T47D from Figure 1. Instead, we inserted some sentence discussing about the data of MCF-7 and T47D data in Discussion part (Results; lines 85~91 & Discussion; lines 245~249; 252~255 in red).

  1. line 275 seems to be wrong style

→ Answer: We corrected line 275 in a right way (lines 297~298 during revision).

  1. page 14 wound healing: How did the authors analyze the images?

→ Answer: To analyze the migrated cells, we appointed randomly selected fields, and the images were taken at same point on the plate at 0, 24 and 48 h after scratching with a sterile pipette tip. Based on the image taken at 0 h, we drew lines and applied to the image taken at 24 and 48 h. Then, we counted the cells migrated into scratched area. To help the readers’ understanding, we added this sentence, “Cells migrated into scratched area were counted.” in the lines 372~373.

  1. page 14 adhesion assay: That seems to be a rather subjective method. The images do not give sufficient space for analysis. It also seems that ECs are not at 90% confluence. This analysis should be repeated using factors which enable discrimination between ECs and breast cancer cells to give trustworthy results.

→ Answer: We have already answered above (please see the answer for Question #3) and added new figures (Fig. 3D~F).

  1. page 14 matrigel invasion assay: how did the authors count the cells?

Answer: As we mentioned in the Methods, MDA-MB-231 or RT-R-MDA-MB-231 cells (2 × 105 cells/500 ml) were added to each upper insert well. After overnight incubation (16 h), we counted the invaded cells which were stained with DAPI in five randomly selected field under a fluorescence microscope. Please see the Method section (4.10. Matrigel invasion assay; lines 389~398) and Figure 2 legend (page 7, lines 139~140).

  1. page 15 Gelatin zymography: If you use supernatant you have to be sure about the number of cells which had been seeded in the first place. How many cells have been seeded prior to the experiment?

→ Answer: To analyze MMP-9 activity with gelatin zymography, 1 x 106 of MDA-MB-231 and RT-R-MDA-MB-231 cells were seeded in 100 mm cell culture dishes. We also added this information in Method section (4.11. Gelatin zymography; lines 400).

  1. In general: how many biological replicates have been used?

→ Answer: Actually, we have repeated three times each experiments (n = 3). During revision, as reviewer 3 suggested, we repeated independent experiments two more to make total n = 5. Accordingly, we corrected each figure legends.

  1. page 15 point 4.15: Were the cells stably transfected or transient? Had transfected clones been selected or did the authors used a heterogeneous culture of undefined transfected cells? It is a huge difference if only one clone is used or different clones in one culture.

→ Answer: 4T1 and RT-R-4T1 cells we used in animal experiments were stably transfected as described in the section of 4.15. (lines 437~440) of the original manuscript as follows, “ESM-1 was overexpressed in 4T1 and RT‑R‑4T1 murine BC by transfection of an ESM-1 plasmid vector (PCMV6-kan/Neo, OriGene Technologies, Inc., Rockville, MD, USA), which contained a neomycin resistance gene for the selection of cells stably expressing ESM-1, in serum-free medium using Lipofectamine 3000 (Thermo Fisher Scientific).”

Following the selection of transfected cell with continuing treatment of neomycin, we selected several single clones and confirmed ESM-1 expression using RT-PCR. We attached the mRNA gel blot images. Please see the attached file.

During revision, to avoid the readers’ confusion and make this point clearer, we modified this sentence into “4T1 and RT‑R‑4T1 murine BC were stably transfected with an ESM-1 plasmid vector (PCMV6-kan/Neo, OriGene Technologies, Inc., Rockville, MD, USA), which contained a neomycin resistance gene for the selection of cells stably expressing ESM-1, in serum-free medium using Lipofectamine 3000 (Thermo Fisher Scientific).” (please see line 437).

  1. page 16 point 4.16: Using standard error of the mean always gives the "herb taste" that the results graphs had to be polished. SD would be a better choice to show visible representative data.

→ Answer: As you suggested, we changed the data presentation as mean ± SD. Please see changed figures and legends.

  1. page 16 Conclusion: please enhance image quality

→ Answer: We have updated Figure 7 (during revision, it has been changed into Figure 7 from Figure 6). During revision, we investigated whether ESM-1 overexpressed in RT-R-MDA-MB-231 cells affects PDK1, PKC and ERK1/2 pathways as key regulators and added some results in Figure 4. Therefore, in new Figure 7, we added PDK1, PKC and ERK1/2 pathways as key regulators to affect transcription factors HIF-1a, STAT-3, and NF-kB.

Reviewer 3 Report

The authors state that ESM-1 overexpression is involved tumorigenesis of radiotherapy-resistant breast cancer cells. Although it is an interesting paper, the manuscript is not enough to explain your concept. The reviewer has provided some major comments below.

Major Point

  1. Because all in vitro studies are not sufficient at N=3, the authors must perform at least N=5 using independent samples. (Fig. 1-5)
  2. The major problem I see is that the study has been performed with a single pair of cell lines (MDA-MB-231 and RT-R-MDA-MB-231). In addition to the results, at a minimum the authors must show several human TNBC cell lines give similar results.
  3. In Figure 1B, the authors had better experiment with HER2-positive cells. (ex. SK-BR-3)
  4. In Table 1, there is a 318.75-fold difference in gene expression between MDA-MB-231 and RT-R-MDA-MB-231, however in Figure 1, it is only about 7-fold difference. Please describe the reason in detail. In addition, also check for the differences in gene expression for the other nine genes using qRT-PCR.
  5. Sagara et al. [Breast Can Res Treat 2017;161:269] showed the extracellular protein level of ESM-1 in MDA-MB-231BR (a brain metastatic variant in MDA-MB-231) was significantly elevated relative to that in MDA-MB-231, consistent with the results of qRT-PCR in these cell lines. However, in Figure 1C, the result shows that the difference is small in ESM-1 protein release relative to ESM-1 gene expression between MDA-MB-231 and RT-R-MDA-MB-231. Please discuss the reason in discussion section in detail.
  6. The authors showed the role of ESM-1 in tumorigenesis in an in vivo mouse model using 4T1 cells. However, in fact, mouse endocan is less glycanated than human endocan, and its biological properties are opposite to human endocan; i.e., glycanated human endocan has protumoral and anti-inflammatoryeffects, but non-glycanated mouse endocan has antitumoral and proinflammatoryeffects [Yassine H et al. Oncotarget 2014;6:2725; Caires NDF et al. Circ Res2015;116:e69]. For these reasons, to explain your concept, the authors must show the role of in tumorigenesis in an in vivo mouse model of mammary fat pad using human TNBC cell lines included MDA-MB-231 and RT-R-MDA-MB-231. In addition, please describe the discussion section about the results of 4T1 model.
  7. In vivo study, please show the ESM1 mRNA expression level in tumor tissues, and plasma level in TNBC cell lines (included MDA-MB-231 and RT-R-MDA-MB-231) xenografts.
  8. In conclusions, the authors describe that ESM-1 may be a target molecule for treating breast cancer, especially TNBC. Therefore, the authors should perform the effect of ESM-1 down-regulation using ex shESM-1.
  9. Why dose ESM-1 up-regulation with radiation resistance? Please describe in discussion section.
  10. Why dose ESM-1 activate transcriptional factors such as HIF-1a/STAT3/NF-kB ? Please describe in discussion section.

Author Response

Comments and Suggestions for Authors

The authors state that ESM-1 overexpression is involved tumorigenesis of radiotherapy-resistant breast cancer cells. Although it is an interesting paper, the manuscript is not enough to explain your concept. The reviewer has provided some major comments below.

Major Point

  1. Because all in vitro studies are not sufficient at N=3, the authors must perform at least N=5 using independent samples. (Fig. 1-5)

→ Answer: Thank you for your comments. According to your suggestion, we performed two more experiments to make n = 5. Please see all figures and legends.

  1. The major problem I see is that the study has been performed with a single pair of cell lines (MDA-MB-231 and RT-R-MDA-MB-231). In addition to the results, at a minimum the authors must show several human TNBC cell lines give similar results.

→ Answer: Thank you for your suggestion. To help the readers’ understanding and make the aim of this study more clearly, we modified the Introduction (please see lines 53 ~ 70). As explained in the Introduction, in our previous study, we established RT‑R-BC cells from TNBC cell (MDA-MB-231) and non-TNBC cells (MCF-7 and T47D), and examined the properties of RT-R-BC cells. We found that RT‑R‑MDA‑MB‑231 cells which is derived from highly metastatic MDA-MB-231 cells showed most radio- and chemo-resistance of tested three cell lines (RT-R-MDA-MB-231, RT-R-MCF-7, RT-R-T47D) and more increased metastatic properties compared to other RT-R-BC cells and BC cells. In addition, we have also reported that RT‑R‑MDA‑MB‑231 cells release higher levels of ATP than other BC cells, and the subsequent activation of P2Y2R by released ATP leads to tumor growth and invasion through inflammasome activation. Therefore, as a further study to overcome radioresistance in BC, in this study, we analyzed gene expression levels between MDA-MB-231 cells a highly metastatic TNBC and RT-R-MDA-MB-231 cells derived from MDA-MB-231 cells based on our previous results. As a representative TNBC, we are using MDA-MB-231 and its RT-R-MDA-MB-231 cells. As cited in the Discussion, ESM-1 is a biomarker of TNBC [19]. So, other TNBC might show similar results with MDA-MB-231 and RT-R-MDA-MB-231. To establish RT-R-BC cells, it takes about 6 months. So, in this study, other RT-R-TNBC cells are not abailavle. We ask this reviewer’s underatanding for this point.

  1. In Figure 1B, the authors had better experiment with HER2-positive cells. (ex. SK-BR-3)

→ Answer: As we explained above, RT‑R‑MDA‑MB‑231 cells which is derived from highly metastatic MDA-MB-231 cells showed most radio- and chemo-resistance of tested three cell lines (RT-R-MDA-MB-231, RT-R-MCF-7, RT-R-T47D) and more increased metastatic properties compared to other RT-R-BC cells and BC cells. In addition, RT‑R‑MDA‑MB‑231 cells released higher levels of ATP than other BC cells, and the subsequent activation of P2Y2R by released ATP leaded to tumor growth and invasion through inflammasome activation. Therefore, as a further study to overcome radioresistance in BC, we were interested of studying in TNBC (ER-/PR-/HER2-).

  1. In Table 1, there is a 318.75-fold difference in gene expression between MDA-MB-231 and RT-R-MDA-MB-231, however in Figure 1, it is only about 7-fold difference. Please describe the reason in detail. In addition, also check for the differences in gene expression for the other nine genes using qRT-PCR.

→ Answer: Gene expression array was performed by the company ‘Ebiogene’ by using the QuantiSeq 3′ mRNA-Seq Service as we described in the Method (Section 4.3. Gene expression array analysis). To confirm the result, we performed RT-PCR. According to the Methodology, gene expression fold would be varied, we think. In our experimental manual for RT-PCR, we got about 7-fold difference in ESM-1 expression between RT-R-MDA-MB-231 and MDA-MB-231. In addition, in this study, we didn’t focus on other 9 genes. Our aim in this study was to investigate the role of ESM-1 the most upregulated gene in RT-R-MDA-MB-231 compared to MDA-MB-231 cells. Please understand this point.

  1. Sagara et al. [Breast Can Res Treat 2017;161:269] showed the extracellular protein level of ESM-1 in MDA-MB-231BR (a brain metastatic variant in MDA-MB-231) was significantly elevated relative to that in MDA-MB-231, consistent with the results of qRT-PCR in these cell lines. However, in Figure 1C, the result shows that the difference is small in ESM-1 protein release relative to ESM-1 gene expression between MDA-MB-231 and RT-R-MDA-MB-231. Please discuss the reason in discussion section in detail.

→ Answer: When we compared the results from Sagara et al., ESM1 gene expression in MDA-MB-231 BR was almost 760 folds higher than that of MDA-MB-231, which is 2 times of our result (320 folds in RT-R-MDA-MB-231/MDA-MB-231). ESM1 protein level observed in Sagara et al was also higher than our result.

The possible difference between two labs could be suggested as follows; First, MDA-MB-231 BR could be more aggressive than RT-R-MDA-MB-231. Second, experimental condition and protocol could be a little different between two labs.

However, the most important common finding is that ESM1 increased in malignant BC cells and play an important role in tumorigenesis of BC.

We discussed this point in the discussion part (please see lines 262~267).

  1. The authors showed the role of ESM-1 in tumorigenesis in an in vivo mouse model using 4T1 cells. However, in fact, mouse endocan is less glycanated than human endocan, and its biological properties are opposite to human endocan; i.e., glycanated human endocan has protumoral and anti-inflammatory effects, but non-glycanated mouse endocan has antitumoral and proinflammatory effects [Yassine H et al. Oncotarget 2014;6:2725; Caires NDF et al. Circ Res2015;116:e69]. For these reasons, to explain your concept, the authors must show the role of in tumorigenesis in an in vivo mouse model of mammary fat pad using human TNBC cell lines included MDA-MB-231 and RT-R-MDA-MB-231. In addition, please describe the discussion section about the results of 4T1 model.

→ Answer: Thank you for your valuable comments. We totally agree with you. In our previous study using mice which were injected with MDA-MB-231 and RT-R-MDA-MB-231, mice developed tumor mass within a few days but decreased tumor mass as a few days went on, because human BC cells couldn’t overcome immune barrier of mice, we think. Even though RT-R-MDA-MB-231 increased tumor growth again in the late phase, we failed to observe the successful metastasis to other organ. When we used SCID mice, we got the similar results.

According to other researchers including Ferrari-Amorotti G et al. performed BC study using both cell lines human BC cells such as MDA-MB-231 and mouse BC cells such as 4T1 cell (Ferrari-Amorotti G et al., 2014; Larive RM et al., 2014; Zhou R et al., 2014). 4T1 cell is an animal model for stage IV human BC as you mentioned, so we expected that it can mimic the biology of MDA-MB-231 which is highly metastatic BC cells. To investigate the role of ESM-1 on tumorigenesis including metastasis in vivo, we made mice xenograft model using 4T1 and RT-R-4T1. Furthermore, 4T1 cell doesn’t express ESM-1, so it is good model to study the role of ESM-1 after ESM-1 transfection, we think. In our experiments, we stably transfected ESM-1 into 4T1 and RT-R-4T1 cells as described in the Methods (4.15 ESM-1 overexpression in 4T1 and RT‑R‑4T1 cells), and injected to mice.

As you suggested, we explained this point briefly in the Discussion (please see lines 305~312).

Besides, to make sure the role of ESM-1 in 4T1 murine BC cells, we performed in vitro colony formation assay and invasion assay with 4T1-EV, 4T1-ESM-1, RT-R-4T1-EV and RT-R-4T1-ESM-1. The results showed that RT-R-4T1 cells exhibited more proliferative (upper panel) and invasive (lower panel) property than 4T1 cells, and ESM-1 overexpression enhanced these abilities in 4T1 murine BC cells. Please see the attached file.

Ferrari-Amorotti G, Chiodoni C, Shen F, Cattelani S, Soliera AR, Manzotti G, Grisendi G, Dominici M, Rivasi F, Colombo MP, Fatatis A, Calabretta B. Suppression of invasion and metastasis of triple-negative breast cancer lines by pharmacological or genetic inhibition of slug activity. Neoplasia. 2014, 16(12):1047-58.

Larive RM, Moriggi G, Menacho-Márquez M, Cañamero M, de Álava E, Alarcón B, Dosil M, Bustelo XR. Contribution of the R-Ras2 GTP-binding protein to primary breast tumorigenesis and late-stage metastatic disease. Nat Commun. 2014, 5:3881.

Zhou R, Xu L, Ye M, Liao M, Du H, Chen H. Formononetin inhibits migration and invasion of MDA-MB-231 and 4T1 breast cancer cells by suppressing MMP-2 and MMP-9 through PI3K/AKT signaling pathways. Horm Metab Res. 2014, 46(11):753-60.

  1. In vivo study, please show the ESM1 mRNA expression level in tumor tissues, and plasma level in TNBC cell lines (included MDA-MB-231 and RT-R-MDA-MB-231) xenografts.

→ Answer: We used up plasma sample for other experiments, so we couldn’t measure the ESM-1 level in plasma. Instead, we confirmed ESM-1 protein level in tumor tissue during revision and added new data in Figure 6. Please understand this situation.

The reason why we didn’t make the mice xenografts model using human TNBC cells (included MDA-MB-231 and RT-R-MDA-MB-231) was explained above. Please see the answer to Question #6.

  1. In conclusions, the authors describe that ESM-1 may be a target molecule for treating breast cancer, especially TNBC. Therefore, the authors should perform the effect of ESM-1 down-regulation using ex shESM-1.

→ Answer: To investigate the role of ESM-1 on tumorigenesis, we did in vitro experiments with ESM-1-knocked down MDA-MB-231 and RT-R-MDA-MB-231 cells using ESM-1 siRNA (Figures 3~5). Furthermore, we performed in vivo experiments with ESM-1-overexpressed 4T1 cells using ESM-1 plasmid (Figure 6) (4T1 cells don’t express ESM1).

  1. Why dose ESM-1 up-regulation with radiation resistance? Please describe in discussion section.

→ Answer: It is not known yet. We also don’t know exactly. In this study, we found that ESM-1 was most upregulated in RT-R-MDA-MB-231 cells and studied the role of ESM-1 in tumorigenesis in RT-R-MDA-MB-231 cells. The topic about ESM-1 up-regulation with radiation resistance will be very interesting in our next project. Thank you for your good suggestion.

  1. Why dose ESM-1 activate transcriptional factors such as HIF-1a/STAT3/NF-kB? Please describe in discussion section.

→ Answer: Our results showed that ESM-overexpressed BC cells such as RT-R-MDA-MB-231 cells showed induced PDK1, PKC and ERK1/2 activation (Figure 4), which were reduced by ESM-1 siRNA.

According to references, PDK1, which is downstream of PI3K and activated by hormones and growth factors, activates several kinase pathways, including the AKT and PKC pathways, which are involved in the regulation of cell growth, proliferation, survival and metabolism [29,30]. PDK1 is overexpressed in human BC models, and PDK1 knockdown abolishes tumorigenesis in xenografts [31]. PKCs are activated by the PI3K and PDK1 pathways and then regulate several genes involved in tumor progression, metastasis and tumorigenesis [32], resulting in the promotion of tumor growth and metastasis [33]. Moreover, PKC increases migration and invasion through the ERK/AP-1 pathway and the production of MMP-9 [34]. The ERK pathway is also involved in tumorigenesis by promoting cell proliferation, survival and angiogenesis, resulting in decreased life expectancy of TNBC patients [35]. Activation of the ERK pathway potentiates the activation of HIF-1α, NF-kB and STAT-3, and these transcription factors are well known to be involved in the tumor growth and progression through the transcriptional regulation of various inflammatory genes, such as adhesion molecules, MMPs, and VEGF [36-40]”.

Therefore, ESM-1-activated PDK1, PKC and ERK1/2 pathway may activate transcriptional factors such as HIF-1a/STAT3/NF-kB which are involved in the transcriptional regulation of various inflammatory genes, such as adhesion molecules, MMPs, and VEGF.

We also included these explanations in the Discussion part (lines 280~293).

Round 2

Reviewer 1 Report

Comments to Jin et al.

This is an original and well-written paper, in line with the journal’s aim and scope, where the main objective is to describe ESM-1 overexpression as involved in increased tumorigenesis of radiotherapy-resistant breast cancer cells.

The research was executed according to currently accepted standards and the paper is interesting to read.

I have appreciated all authors remarks to my main requests regarding their work. They have modified the paper according to my main concerns and I think that the entire manuscript was really improved.

An only remark following described remain. I have previously requested an explanation regarding the reason why mice were not xenotransplanted with the MDA-MB-231 cell line. Even if I understand and accept your choice linked to method troubleshooting, I strongly suggest to overcome your troubles in the future. In fact, as according to my previous experience and literature data, is possible to generate MDA-MB-231 xenograft mice using BALB/c nude mice that lack a thymus and thus unable to produce T-cells and are therefore immunodeficient.

Anyhow, in my opinion, the manuscript is now ready for publication in the present form.

Reviewer 2 Report

Dear Authors,

thank you for revising the manuscript as suggested. From my point of view there are no further changes needed despite Figure 7 needs quality improvement.

Congratulations

Reviewer 3 Report

OK.

In the future, please try in vivo experiment again using NSG mice.